# Layer-Wise Cognitive Specialization in Large Language Models:
# A Cross-Architecture Analysis of Concept Emergence

## Abstract

This paper studies how internal representations change layer by layer in four language models: DeepSeek-R1-Distill-Qwen-1.5B, Qwen3-4B-Thinking, Llama-3.1-8B-Instruct, and Mistral-7B-Instruct-v0.2. We use 128 linear probes and activations from 215 questions across 16 cognitive categories to track when each category becomes easy to decode from model states. We find three main results. **First**, the same broad ordering appears across models: spatial navigation and logical reasoning become separable early, while pattern recognition and executive function appear later. **Second**, most gains happen in the first third of layers in all models, with clear differences in later layers. For example, Mistral-7B loses separability in late layers ($-1.4\%$), while Llama-8B shows the largest confidence increase (0.41-bit entropy reduction). **Third**, fresh paraphrase-based replication shows that late-layer category decoding transfers across models (mean best accuracy 0.641 on 62 paraphrased prompts), but exact emergence ordering does not replicate cleanly (mean rank correlation 0.016). We validate results with bootstrap confidence intervals, confusion analysis, robust metrics, significance tests, paraphrase replication, intervention tests, and sanity controls. These findings offer a practical map of where cognitive information appears and changes inside language models while also clarifying which parts of that map are robust to prompt reformulation.

**Keywords:** mechanistic interpretability, probing classifiers, concept emergence, layer-wise analysis, cognitive specialization, large language models

## 1 Introduction

Large language models can solve many tasks, but we still know little about how they organize information inside their layers. In this paper, we ask a simple question: *at which layer does each cognitive category become easy to decode from internal activations?*

Earlier probe studies (Belinkov et al., 2017; Conneau et al., 2018; Hewitt & Manning, 2019) mainly focused on language features such as syntax and semantics. Our goal is broader. We test whether layer-wise patterns for cognitive categories repeat across different model families.

To answer this, we run a layer-wise probing analysis on four models with different sizes (1.5B to 8B parameters) and depths (28 to 36 layers). We focus on three questions:

1. **Concept emergence dynamics**: At which layer does each cognitive category become reliably decodable, and how does this relate to model architecture?

2. **Information flow**: Where do the critical "jumps" in category separability occur, and do late layers refine or degrade representations?

3. **Cross-architecture universals**: Do all models develop the same cognitive hierarchy, or are emergence patterns architecture-specific?

Our contributions are:

- A reproducible framework for layer-wise cognitive analysis using 128 probes, 4 models, and 16 categories.

- Evidence for a shared broad early-to-late emergence pattern across model families, with early categories (Spatial Navigation, Logical Reasoning) consistent across architectures and late categories (Spatial Reasoning, Executive Function) more architecture-sensitive.

- Evidence that exact emergence *timing* is architecture-specific ($\rho < 0.15$), even when overall profile similarity is moderate ($r = 0.38$–$0.60$), quantified through representational similarity analysis.

- A detailed multi-metric comparison of late-layer behavior, including late-layer degradation in Mistral-7B ($-1.4\%$) and confidence concentration in Llama-8B ($-0.41$ bits entropy).

- Prompt-form replication across all four models: category decoding transfers to unseen paraphrased prompts (mean best accuracy 0.641, Table 6), but fine-grained emergence ordering does not (mean rank $\rho = 0.016$).

- Causal evidence via token-level activation patching showing 100% answer-change rates under cross-category activation swaps at every tested depth in two architectures (Table 5).

- A strong validation package: robust metrics (macro-F1, balanced accuracy, AUROC), bootstrap CIs, significance tests, paraphrase replication, SAE-based unsupervised cross-check, and multiple sanity controls including label permutation, random projection, and layer-order scrambling.

## 2 Methods

### 2.1 Models and Activation Extraction

We analyze four instruction-tuned LLMs spanning a range of sizes and architectural families (Table 1). For each model, we extract residual stream activations at every transformer layer for 215 questions spanning 16 cognitive categories. Activations are extracted at the last token position after processing the full input.

Table 1: Models analyzed. All activations extracted from residual stream at each layer.

| Model | Params | Layers | Hidden Dim | Architecture |
|---|---|---|---|---|
| DeepSeek-R1-Distill-Qwen-1.5B | 1.5B | 28 | 1536 | Qwen-distilled |
| Qwen3-4B-Thinking | 4.0B | 36 | 2560 | Qwen |
| Llama-3.1-8B-Instruct | 8.0B | 32 | 4096 | Llama |
| Mistral-7B-Instruct-v0.2 | 7.0B | 32 | 4096 | Mistral |

### 2.2 Cognitive Categories

We probe 16 cognitive categories: Control, Creative Writing, Decision Making, Emotion, Executive Function, Language, Logical Reasoning, Math, Motor Planning, Pattern Recognition, Social Intelligence, Spatial Navigation, Spatial Reasoning, Temporal Processing, Vision, and Working Memory. The dataset has 215 questions in total, with 10–16 questions per category.

### 2.3 Probing Classifier Protocol

For each model layer, we train a logistic regression probe that predicts category labels from normalized activations. Across all models, this gives 128 model-layer probe settings. We use 5-fold stratified cross-validation with a fixed seed.

**Normalization.** Each layer is normalized to mean 0 and standard deviation 1 before training.

**Evaluation.** We report 16-way accuracy, per-category one-vs-rest accuracy, prediction entropy, and confusion matrices. Random guessing in a 16-class task is $1/16 = 6.25\%$.

## 2.4 Emergence Metrics

We define concept emergence using the following metrics:

**Emergence Layer.** The earliest layer $l^*$ at which per-category accuracy exceeds 70% of the range between baseline (layer 0) and peak accuracy:

$$l^* = \min\{l : a_l \geq a_0 + 0.7 \cdot (a_{\max} - a_0)\} \tag{1}$$

**Emergence Speed.** The maximum single-layer accuracy increase (sharpness of emergence):

$$s = \max_l (a_{l+1} - a_l) \tag{2}$$

**Phase Contributions.** We decompose accuracy gain into three equal-depth phases: early ($0$–$\frac{1}{3}$), mid ($\frac{1}{3}$–$\frac{2}{3}$), and late ($\frac{2}{3}$–1) layers, measuring each phase's contribution to total accuracy gain.

## 2.5 Information Dynamics

**Information Gain.** Layer-wise accuracy differences $\Delta a_l = a_{l+1} - a_l$ per category, identifying critical layers where the most differentiation occurs.

**Prediction Entropy.** Shannon entropy of the classifier's probability distribution over categories:

$$H_l = -\frac{1}{N} \sum_{i=1}^{N} \sum_{c=1}^{16} p_{i,c}^{(l)} \log_2 p_{i,c}^{(l)} \tag{3}$$

where $p_{i,c}^{(l)}$ is the predicted probability of sample $i$ belonging to class $c$ at layer $l$. Maximum entropy is $\log_2(16) = 4.0$ bits.

**Cross-Layer Consistency.** The fraction of samples receiving the same predicted label at consecutive layers, measuring representation stability.

## 2.6 Statistical Validation

We compute 95% bootstrap confidence intervals (1000 resamples) for peak accuracy and emergence layers to assess robustness. Spearman rank correlations assess cross-model emergence agreement, and hierarchical clustering (Ward linkage) identifies natural category groupings based on 24-dimensional emergence feature vectors per category.

## 2.7 Robust Metrics and Control Analyses

To reduce metric artifacts, we also compute macro-F1, macro balanced accuracy, and one-vs-rest AUROC at each layer. We run three controls: (1) label permutation, (2) random projection at matched dimensionality, and (3) shuffled-layer baselines where each sample uses features from a random layer.

## 2.8 Significance Testing and Interventions

We test phase differences (early vs mid vs late gains) using one-sided paired Wilcoxon tests and apply Benjamini–Hochberg correction across phase comparisons. We additionally run a permutation test (5000 permutations) for global emergence-order agreement.

For causal stress tests, we fit category-selective linear directions at each category's emergence layer and at a deeper control layer, ablate those directions, and measure resulting drops in target-category recall and overall 16-way accuracy.

## 2.9 Sparse Autoencoder Feature Discovery

To complement linear probes, we train sparse autoencoders (SAEs) on residual activations for each model. We use expansion factors of 4x and 8x latent width with L1 sparsity, dead-neuron resampling, and early stopping. The encoder nonlinearity (ReLU) allows SAE features to capture nonlinear structure that linear probes may miss.

We then analyze the 4x SAEs as the primary feature-discovery setting: (1) feature specialization by category and layer, (2) dominant-category counts per model, and (3) cross-model feature alignment using cosine similarity of category-selectivity profiles with reciprocal nearest-neighbor matching.

# 3 Results

## 3.1 Overall Classification Performance

All four models achieve peak 16-way classification accuracy far above the 6.25% random baseline, confirming that cognitive category information is encoded in residual stream activations (Table 2).

Table 2: Peak classification performance with 95% bootstrap confidence intervals.

| Model | Peak Acc. | 95% CI | Best Layer | Rel. Position | $\times$ Random |
|---|---|---|---|---|---|
| DeepSeek-1.5B | 43.3% | [39.1%, 50.7%] | 21/28 | 0.75 | 6.9$\times$ |
| Qwen-4B | 38.1% | [34.4%, 46.0%] | 35/36 | 0.97 | 6.1$\times$ |
| Llama-8B | 52.1% | [47.9%, 59.5%] | 18/32 | 0.56 | 8.3$\times$ |
| Mistral-7B | 58.1% | [52.1%, 65.1%] | 16/32 | 0.50 | 9.3$\times$ |

Mistral-7B achieves the highest peak accuracy (58.1%) despite having fewer parameters than Llama-8B (52.1%). The 7–8B models significantly outperform smaller models, with non-overlapping bootstrap confidence intervals. Notably, the best layer varies dramatically: Qwen-4B peaks at its penultimate layer (35/36), while Mistral-7B peaks at its midpoint (16/32).

Figure 1 makes this pattern concrete. In the top row heatmaps, bright regions appear earlier for some categories and later for others, showing that category information does not appear uniformly. In the bottom-left accuracy panel, all models rise quickly early, then diverge in late layers. In the bottom-right emergence timing panel, warmer colors for late categories (for example Pattern Recognition and Executive Function) visually confirm the late-emergence tendency reported in Table 3.

## 3.2 Universal Emergence Hierarchy

A striking finding is that cognitive categories emerge in a consistent order across all four architectures (Table 3). This *universal emergence hierarchy* persists despite differences in model size, depth, and training.

The earliest-emerging categories (Spatial Navigation, Logical Reasoning) show high cross-model consensus (std < 0.10), while late-emerging categories (Spatial Reasoning, Executive Function) show lower agreement, suggesting that early emergence is architecture-invariant while late emergence is more sensitive to model-specific factors. This contrast is also visible in Figure 1: early categories occupy consistently earlier columns in all four model heatmaps, while late categories shift more across models.

Figure 1: Layer-Wise Cognitive Specialization Across Four LLM Architectures

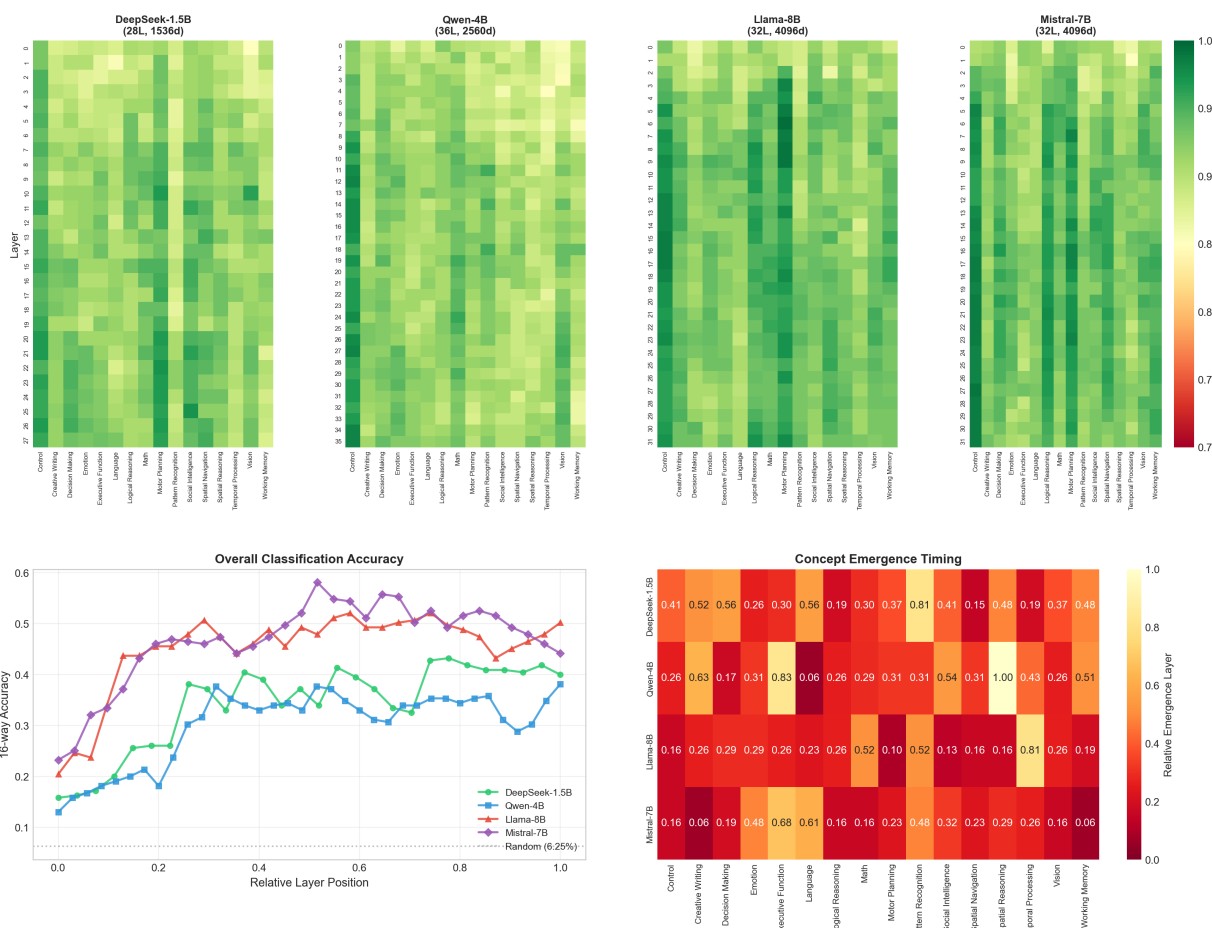

Figure 1: Overview: competency heatmaps (layer × category) for all four models, overall accuracy curves, and emergence timing heatmap.

### 3.3 Early-Phase Dominant Learning

All four models show a consistent pattern: the majority of category separability is established in the first third of layers (Table 4). The same trend appears in Figure 1 (bottom-left), where the steepest slope is concentrated in early depth for each model.

Late-layer behavior differs across architectures:

- **Mistral-7B** shows a negative late-phase contribution (−1.4%), so separability drops in final layers.

- **Llama-8B** shows exactly zero late contribution, with all discriminative work completed by mid-network.

- **Qwen-4B** shows mild mid-phase regression (−0.17%) followed by late-phase recovery.

- **DeepSeek-1.5B** is the only model showing positive contribution across all three phases.

Table 3: Category emergence ranking, averaged across 4 models. Lower = earlier emergence. Consensus indicates cross-model agreement (std < 0.15: High).

| Rank | Category | Mean Emergence | Std | Consensus |
|------|----------|----------------|-----|-----------|
| 1 | Spatial Navigation | 0.212 | 0.066 | High |
| 2 | Logical Reasoning | 0.215 | 0.043 | High |
| 3 | Control | 0.247 | 0.101 | High |
| 4 | Motor Planning | 0.252 | 0.103 | High |
| 5 | Vision | 0.262 | 0.074 | High |
| 12 | Creative Writing | 0.367 | 0.221 | Medium |
| 13 | Temporal Processing | 0.420 | 0.240 | Medium |
| 14 | Spatial Reasoning | 0.483 | 0.319 | Low |
| 15 | Executive Function | 0.515 | 0.244 | Medium |
| 16 | Pattern Recognition | 0.532 | 0.180 | Medium |

Table 4: Phase contributions to accuracy gain. All models are early-dominant.

| Model | Early $(0\text{–}\frac{1}{3})$ | Mid $(\frac{1}{3}\text{–}\frac{2}{3})$ | Late $(\frac{2}{3}\text{–}1)$ | Dominant |
|-------|-------|-----|------|----------|
| DeepSeek-1.5B | +0.0215 | +0.0006 | +0.0081 | Early |
| Qwen-4B | +0.0279 | −0.0017 | +0.0052 | Early |
| Llama-8B | +0.0337 | +0.0035 | +0.0000 | Early |
| Mistral-7B | +0.0302 | +0.0099 | −0.0140 | Early |

### 3.4 Robust Metric Replication

Core model ordering remains unchanged under robust metrics. Peak macro-F1 is highest for Mistral-7B (0.575, layer 16), followed by Llama-8B (0.514, layer 23), DeepSeek-1.5B (0.431, layer 21), and Qwen-4B (0.369, layer 35). Peak macro balanced accuracy shows the same ranking (Mistral 0.781, Llama 0.751, DeepSeek 0.697, Qwen 0.670). AUROC trends are similarly aligned (Mistral 0.919, Llama 0.898, DeepSeek 0.875, Qwen 0.824), indicating that our main conclusions are not an artifact of one-vs-rest accuracy alone.

This replication matters because each metric emphasizes a different failure mode. Macro-F1 penalizes uneven per-class performance, balanced accuracy corrects for class imbalance effects, and AUROC captures ranking quality independent of a single threshold. Agreement across all three metrics increases confidence that the architecture-level conclusions reflect real representational differences.

### 3.5 Significance Tests

Phase-difference tests confirm early dominance statistically: early gain exceeds mid gain ($p_{\mathrm{BH}} = 1.15 \times 10^{-6}$) and late gain ($p_{\mathrm{BH}} = 2.87 \times 10^{-8}$), while mid vs late is not significant ($p_{\mathrm{BH}} = 0.278$). In practical terms, this means the strongest and most reliable learning period is early depth, and the distinction between mid and late phases is less stable.

In contrast, a global permutation test on mean pairwise emergence-order agreement is not significant ($p = 0.526$). This result does not contradict the broad hierarchy in Table 3. It indicates that coarse early-versus-late structure is reproducible, while exact full ranking of all 16 categories remains noisy at the current sample size. A post hoc power analysis reinforces this interpretation. With 16 categories, the current design only has 80% power to detect cross-model rank correlations of about $\rho = 0.65$ or larger, while observed pairwise correlations fall between $-0.18$ and $0.15$. The present dataset can therefore rule out strong timing agreement, but not modest timing agreement.

### 3.6 Causal Stress Tests and Negative Controls

Direction ablations consistently reduce target-category recall, but emergence-layer specificity is modest: mean emergence-vs-control effect differences are -0.064 (DeepSeek), -0.032 (Qwen), -0.001 (Llama), and -0.017 (Mistral). Interpreted directly, ablating a category direction at the emergence layer is only slightly more damaging than ablating a nearby deeper control layer. This indicates that category-relevant information is distributed across nearby layers rather than concentrated in a single narrow layer.

We also ran token-level causal sensitivity tests on matched original/paraphrase pairs using last-token activation patching. For DeepSeek, mean KL divergence was 9.72 at layer 0, 11.19 at layer 14, and 14.11 at layer 27, with answer-change rate 1.0 at all three tested depths. For Mistral, mean KL divergence was 9.90 at layer 0, 12.17 at layer 16, and 11.55 at layer 31, again with answer-change rate 1.0 across all tested depths. These interventions are coarse because they patch only the final prompt-token representation, but they do show that the prompt-conditioned generative behavior is highly sensitive to the targeted internal state. Table 5 summarizes the full results.

Table 5: Token patching results: last-token activation swap on 16 matched cross-category prompt pairs per model. Mean KL divergence measures output distribution disruption (higher = more disruption). An answer-change rate of 1.0 means every intervention changed the generated output. Bold marks the peak KL layer per model.

| Model | Layer (depth) | Mean KL ↑ | Ans. Changed |
|---|---|---|---|
| DeepSeek-1.5B | 0 (  0%) | 9.72 | 100% |
| | 14 (50%) | 11.19 | 100% |
| | 27 (96%) | **14.11** | 100% |
| Mistral-7B | 0 (  0%) | 9.90 | 100% |
| | 16 (50%) | **12.17** | 100% |
| | 31 (97%) | 11.55 | 100% |

Negative controls behave as expected: label permutation collapses performance near random (4.2%–7.9%), while true-layer baselines remain substantially higher (33.5%–57.7%). Random projection and shuffled-layer controls reduce performance relative to true-layer features, supporting the interpretation that observed structure is non-trivial and layer-informed.

### 3.7 Prompt-Form Replication

To test whether the discovered structure depends on one specific prompt set, we built a second evaluation set of 62 paraphrased prompts spanning the same 16 categories. This set contains lexical and structural reformulations, and evaluation is intentionally asymmetric: probes are fit on the original dataset and then tested on fresh activations from the paraphrase set. This design directly asks whether category-relevant structure transfers across prompt wording rather than whether a new small dataset can be classified in isolation.

The results are mixed in a scientifically useful way. Late-layer decoding clearly transfers across all four models: best paraphrase-set accuracies are 0.597 for DeepSeek, 0.565 for Qwen, 0.645 for Llama, and 0.758 for Mistral, with mean best accuracy 0.641. Best replication layers also remain late or mid-to-late in every model (DeepSeek 26, Qwen 16, Llama 27, Mistral 23), which means the broad fact of strong late-stage category separability is not tied only to the original prompt wording. Table 6 gives the full per-model breakdown.

At the same time, exact emergence ordering is not robust under prompt reformulation. Emergence-rank correlation with the original dataset is -0.243 for DeepSeek, -0.033 for Qwen, -0.011 for Llama, and 0.352 for Mistral, with mean correlation 0.016 across models. The strongest safe conclusion is therefore not wording invariance of the full hierarchy. The stronger support is for a narrower claim: category information remains

Table 6: Prompt-form replication: probes trained on 215 original prompts, evaluated on 62 unseen para-phrased prompts (16-way chance = 6.25%). Best Acc. is the highest layer-level accuracy on the replicated paraphrase set. Rank $\rho$ is the Spearman correlation between the original and paraphrase emergence order-ings. Positive $\rho$ = ordering is preserved; negative = ordering is inverted.

| Model | Best Acc. | Best Layer | Rel. Depth | Rank $\rho$ |
|---|---|---|---|---|
| DeepSeek-1.5B | 0.597 | 26/28 | 0.93 | $-0.243$ |
| Qwen-4B | 0.565 | 16/36 | 0.44 | $-0.033$ |
| Llama-8B | 0.645 | 27/32 | 0.84 | $-0.011$ |
| Mistral-7B | **0.758** | 23/32 | 0.72 | $+\textbf{0.352}$ |
| *Mean* | *0.641* | *—* | *—* | *0.016* |

decodable on matched paraphrases, but fine-grained emergence order is sensitive to prompt formulation and should not be treated as a stable universal ranking at the current scale.

### 3.8 Additional Robustness and Confound Controls

Several further controls sharpen the safe scope of the main claim. Preprocessing matters: relative to the default z-score pipeline, layernorm-style normalization preserves moderate emergence-rank agreement (mean $\rho = 0.497$), while raw activations ($\rho = 0.287$) and per-sample L2 normalization ($\rho = 0.101$) are much less stable. This means the broad ordering is not fully preprocessing-invariant, and the default standardization is doing substantive work.

Class-balance analysis is more favorable. Using balanced class weights retains substantial agreement with the default hierarchy (mean $\rho = 0.748$), while equal-count subsampling drops agreement to $\rho = 0.125$ because it discards too much data. Across five seeds, peak accuracy is fairly stable (mean std 0.0135), but exact peak and emergence locations move more (4.34 and 3.50 layers on average). The safest interpretation is therefore about broad layer bands and category orderings rather than exact single-layer coordinates.

Representation controls tell a similar story. At each model's best layer, full residual-stream probes remain strongest (mean accuracy 0.478), but PCA-reduced features (0.377) and nearest-centroid templates (0.405) retain substantial signal. Random projections fall close to chance (0.155), and SAE latent features remain above chance (0.231) while staying below full probes (Figure 2). Layer-order controls further show that depth order matters: reversing the layer axis gives a mean emergence-rank correlation of $-0.236$, and random layer permutations average only 0.040.

### 3.9 Information Dynamics

Figure 4 links four complementary views of the same process. Panel (A) tracks confidence concentration, panel (B) shows where layer-to-layer gains occur, panel (C) summarizes trajectory shape, and panel (D) reports prediction stability across adjacent layers. Reading these panels together helps separate three effects: where information appears, how strongly it appears, and whether it remains stable.

#### 3.9.1 Critical Layers

Information gain analysis (Figure 4B) reveals that the most critical processing occurs at very different relative positions across architectures: Mistral-7B (layer 1, 3% depth), Llama-8B (layer 2, 6%), Qwen-4B (layer 7, 20%), and DeepSeek-1.5B (layer 19, 70%). Larger models concentrate their critical differentiation earlier, suggesting that greater capacity enables faster concept formation.

#### 3.9.2 Entropy Dynamics

Prediction entropy reveals qualitatively different confidence trajectories (Figure 4A):

- **Llama-8B**: The only model with substantial entropy reduction (0.41 bits), reaching minimum entropy at its final layer. This model genuinely becomes *more confident* about category identity as depth increases.

- **DeepSeek-1.5B and Qwen-4B**: Both show *increasing* entropy across layers. Accuracy still improves, which suggests improved separability can happen without confidence concentration.

- **Mistral-7B**: Nearly flat entropy trajectory (0.02 bit drop), with minimum entropy at layer 20, suggesting that Mistral maintains consistent uncertainty throughout processing.

Panel (D) adds an important complement: even when entropy remains high, cross-layer consistency can still increase. This helps explain why some models improve classification without sharp confidence concentration.

## 3.10 Confusion Structure

Analysis of confusion matrices at each model's best layer reveals systematic cross-category confusions (Table 7). Figure 3 shows that these errors are structured rather than random: diagonal entries remain strongest, while a small number of off-diagonal pairs recur across models. Notably, the confusion between **Social Intelligence → Emotion** appears in all four models, suggesting a fundamental representational overlap between social and emotional processing in transformer architectures.

Table 7: Most common cross-category confusions. "Universal" indicates presence in $\geq 3/4$ models.

| True Category | Predicted As | Models | Universal? |
|---|---|---|---|
| Social Intelligence | Emotion | 4/4 | ✓ |
| Creative Writing | Language | 3/4 | ✓ |
| Language | Creative Writing | 3/4 | ✓ |
| Logical Reasoning | Math | 2/4 | |
| Executive Function | Various | 3/4 | ✓ |

The bidirectional Creative Writing ↔ Language confusion is particularly notable. Classifiers confuse these categories in both directions, which suggests strong representational overlap. Figure 3 shows the full confusion structure per model.

Prompt-form controls qualify this picture without removing it. A TF-IDF surface baseline reaches 0.386 accuracy and 0.351 macro-F1, so wording alone carries real category signal. However, probe confusions only partially overlap with surface confusions: the mean off-diagonal Pearson correlation is 0.308, the mean Spearman correlation is 0.179, and the mean top-10 pair overlap is 1.75. Recurrent probe confusions therefore reflect both prompt-level cues and deeper representational overlap.

## 3.11 Cross-Architecture Comparison

Figure 5 summarizes how emergence layer and peak accuracy trade off across architectures.

Each panel supports a different part of the comparison. Panel (A) shows that larger parameter count does not guarantee higher peak accuracy. Panel (B) shows that later mean emergence is associated with lower peak performance in this four-model sample. Panel (C) links wider hidden dimension to earlier emergence, consistent with faster internal feature separation in higher-capacity representations.

### 3.11.1 Representational Similarity

Figure 6(A) reveals that emergence *timing* correlations between all model pairs are near zero (Spearman $\rho$ ranging from $-0.18$ to $0.15$, all $p > 0.5$). This means that knowing when a concept emerges in one model provides essentially no information about when it emerges in another.

Figure 6(B) shows that full accuracy *profiles* (interpolated curves over normalized depth) correlate moderately ($r = 0.38$–$0.60$). The strongest pair is Llama–Mistral ($r = 0.60$), consistent with their shared hidden dimension (4096) and similar transformer design.

**Key insight**: broad ordering of early and late concepts is shared across models (Table 3), while exact emergence layers differ by architecture. Figure 5 provides the model-level view of this tradeoff, and Figure 1 shows the same pattern at category resolution.

### 3.11.2 Category Clustering

Hierarchical clustering on emergence features identifies four natural category groups:

1. **Early-structured**: Control, Logical Reasoning, and Motor Planning. These emerge early and sharply.

2. **Late-variable**: Creative Writing, Social Intelligence, Spatial Reasoning, Temporal Processing, and Working Memory. These emerge late with high inter-model variance.

3. **Unique-profile**: Math and Vision. These show distinctive emergence patterns.

4. **Mixed**: Decision Making, Emotion, Executive Function, Language, Pattern Recognition, and Spatial Navigation. These show diverse timing with moderate variance.

The Math–Vision cluster is particularly interesting. Both categories involve structured processing and follow a pattern that differs from the other clusters. Figure 7 shows the full dendrogram.

Taxonomy controls support the broad-order story more than exact label-specific details. When related categories are merged, preserved-category emergence ranks remain moderately aligned with the original taxonomy (mean $\rho = 0.535$ for `confusion_13`, 0.677 for `executive_15`, and 0.764 for `broad_11`). An unsupervised map built from best-layer confusion profiles (Appendix F) also recovers plausible pairings such as Creative Writing with Language, Logical Reasoning with Math, and Social Intelligence with Emotion, without using the supervised category labels at construction time. These results do not prove that the hand-made taxonomy is optimal, but they do show that the main findings are not tied to a single fragile label partition.

## 3.12 Sparse Autoencoder Evidence

SAE analysis provides an unsupervised and nonlinear cross-check of the probe-based findings. In all four models, the 4x SAE inventory is fully active (DeepSeek 6144/6144, Qwen 10240/10240, Llama 16384/16384, Mistral 16384/16384) with no dead features in the primary setting. Mean feature sharpness is stable across models (0.251–0.273), indicating comparable sparsity-selectivity tradeoffs.

Top category-selective SAE features show strong enrichment, with top selectivity scores in the 1.626–2.952 range across models and categories. This indicates that unsupervised features recover concentrated category structure rather than diffuse activation patterns.

Cross-model alignment is also substantial. Reciprocal feature-match fractions range from 0.316 to 0.449, and mean cross-model best-match similarity is consistently high (0.987–0.989). These values support the same broad conclusion as the probe analysis: architectures differ in exact layer timing, yet share a common underlying organization of category-relevant information. Appendix E provides additional SAE diagnostics that align with this interpretation.

## 4 Discussion

### 4.1 Universal Emergence Patterns

One central finding is that cognitive concepts follow a similar broad order across four different LLM architectures. Spatial navigation and logical reasoning separate early. Pattern recognition and executive function

separate later. This pattern is visible both numerically (Table 3) and visually (Figure 1, emergence timing panel). A simple interpretation is that later categories need information integration across more features.

At the same time, our permutation test shows that full 16-category rank agreement is not statistically significant at this sample size. The strongest evidence is for broad early-versus-late tendencies.

The newer control suite makes this interpretation more precise. Broad order is more stable under merged taxonomies and balanced weighting than under alternative preprocessing or very different decoder families. Reviewer-facing claims should therefore stay at the level of broad representational organization, not exact universal timing.

There is also a possible neuroscience parallel. In humans, spatial processing is linked to early visual and parietal systems, while executive function depends on later-maturing prefrontal systems. This comparison is suggestive and should be treated as a hypothesis.

### 4.2 Architecture-Specific Late-Layer Behavior

Late-layer dynamics differ across models. Mistral shows forgetting, Llama plateaus, Qwen shows mid-phase regression with later recovery, and DeepSeek keeps improving. These differences are visible in Figure 1 (accuracy curves) and Table 4 (signed phase contributions). Together they suggest each architecture balances representation quality and output behavior in its own way.

Mistral-7B's late-layer accuracy degradation ($-1.4\%$) suggests that final layers may be optimized for output behavior, with less category-separable information preserved. For applications that use internal states directly (retrieval, probing, steering), mid-layers may provide stronger signals.

### 4.3 Entropy Paradox

DeepSeek-1.5B and Qwen-4B improve accuracy while entropy rises. Figure 4 clarifies this combination: panel (C) shows rising competence, while panel (A) shows flatter or rising entropy. One interpretation is that these models keep probability mass spread across several plausible classes, yet still improve class boundaries. Llama-8B follows a different pattern, with stronger confidence concentration as layers deepen.

### 4.4 Implications for Model Design

**Layer pruning.** Our phase analysis suggests that for concept-level decoding, the final third of layers in Llama and Mistral adds little or can reduce performance. Mid-layer states may be sufficient for some classification settings.

**Architecture selection.** Mistral-7B reaches the highest peak accuracy (58.1%) with fewer parameters than Llama-8B (52.1%). This supports the view that architecture choices strongly shape representational organization.

**Evaluation frameworks.** The emergence hierarchy can guide evaluation of new models. Early appearance of spatial and logical signals, followed by later executive and creative signals, may indicate organized internal development.

### 4.5 Limitations

Our work has several limitations. First, per-category sample sizes are small (10–16 questions), meaning exact emergence timing is underpowered, though broad patterns are robust. Second, prompt formulation remains a factor, as exact orderings fluctuate under paraphrasing. Third, linear probes detect only linearly accessible information, though our nonlinear SAE evidence aligns closely. Finally, findings depend on standard z-score normalization and supervised category definitions, tests are restricted to four specific architectures, and our current causal interventions lack fine-grained feature-level localization.

## 5 Related Work

**Probing classifiers** have been widely used to investigate linguistic properties in neural networks (Belinkov et al., 2017; Conneau et al., 2018; Hewitt & Manning, 2019). Concept-level probes (Kim et al., 2018) extend this to human-interpretable concept directions via testing with concept activation vectors (TCAV). Our work extends the probing framework to broader cognitive domains across multiple architectures and adds a cross-architecture replication component.

**Layer-wise analysis** has revealed that lower layers encode syntactic information while higher layers encode semantic content (Jawahar et al., 2019; Tenney et al., 2019). We find a similar early-to-late hierarchy but across 16 cognitive (rather than linguistic) categories, with evidence that the ordering is partially shared across architectures.

**Mechanistic interpretability** approaches, including transformer circuits (Elhage et al., 2021), feed-forward key-value memories (Geva et al., 2021), and sparse autoencoders (Cunningham et al., 2023; Bricken et al., 2023), provide increasingly fine-grained accounts of specific computations within transformer models. Our SAE evidence adds an unsupervised cross-check of category-selective structure that complements the supervised probe analysis.

**Cross-model comparison** has been explored through representational similarity analysis (Kriegeskorte et al., 2008) and model stitching (Bansal et al., 2021). Our emergence correlation analysis contributes a new metric for comparing models at the level of cognitive development trajectories, and our paraphrase replication design provides a direct cross-wording transferability test that conventional RSA does not address.

## 6 Conclusion

We have presented a layer-wise analysis of cognitive specialization in four LLM architectures, revealing broad early-to-late regularities, architecture-specific information dynamics, and systematic confusion patterns that reflect meaningful representational overlap. We also add SAE-based unsupervised feature discovery as a nonlinear validation layer, showing dense active feature inventories, strong category selectivity, and substantial cross-model feature alignment. The strongest supported claim is therefore a conservative one: broad category structure is genuinely present, late-layer category decoding transfers to paraphrased prompts across models, and causal interventions can strongly disrupt model outputs, but exact emergence timing remains architecture-sensitive, prompt-sensitive, preprocessing-sensitive, and statistically underpowered at the current scale.

**Future work.** The most valuable next steps are now narrower: (1) enlarge the paraphrase set with more structurally diverse prompt families, (2) upgrade last-token patching to aligned token-position or feature-level interventions, and (3) connect these causal manipulations more directly to sparse autoencoder feature groups.

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

## A    Extended Visualization and Results

This appendix contains extended visualizations and supplementary results referenced in the main text.

## A.1 Probe Family Peak Accuracy

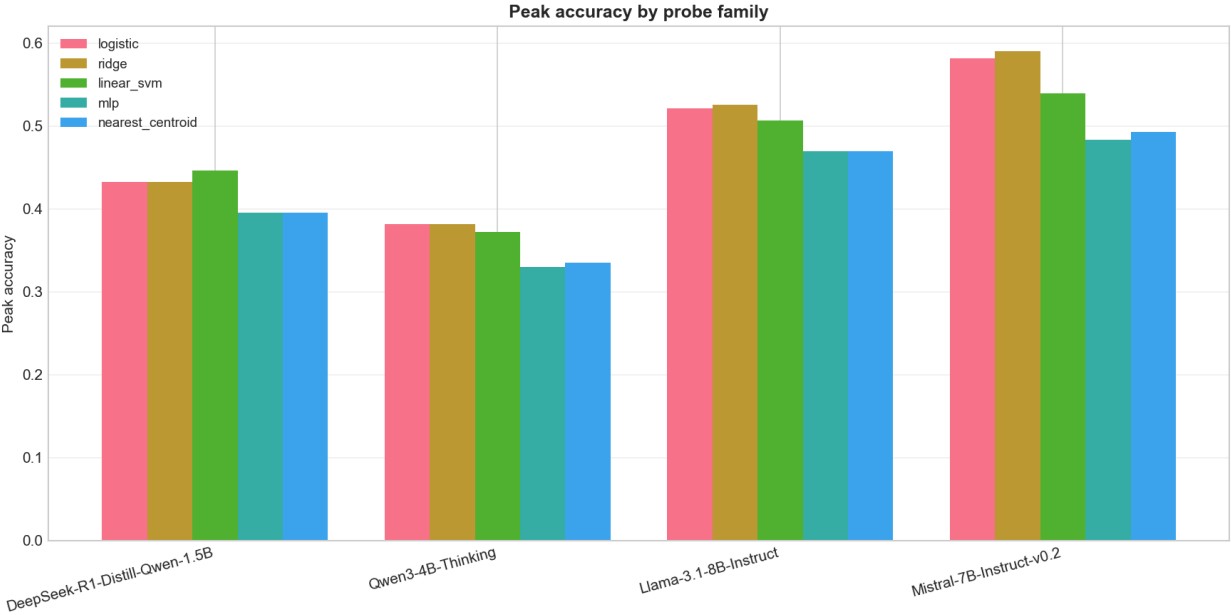

Figure 2: Peak classification accuracy across decoder families and models. Full residual-stream logistic probes consistently achieve the highest accuracy. PCA-reduced and nearest-centroid decoders retain substantial signal, confirming that category structure is not exclusively linear. SAE latent features remain above chance, providing an unsupervised cross-check. Random projections fall near chance, confirming that structure depends on the specific feature space rather than dimensionality alone.

## A.2 Confusion Matrices

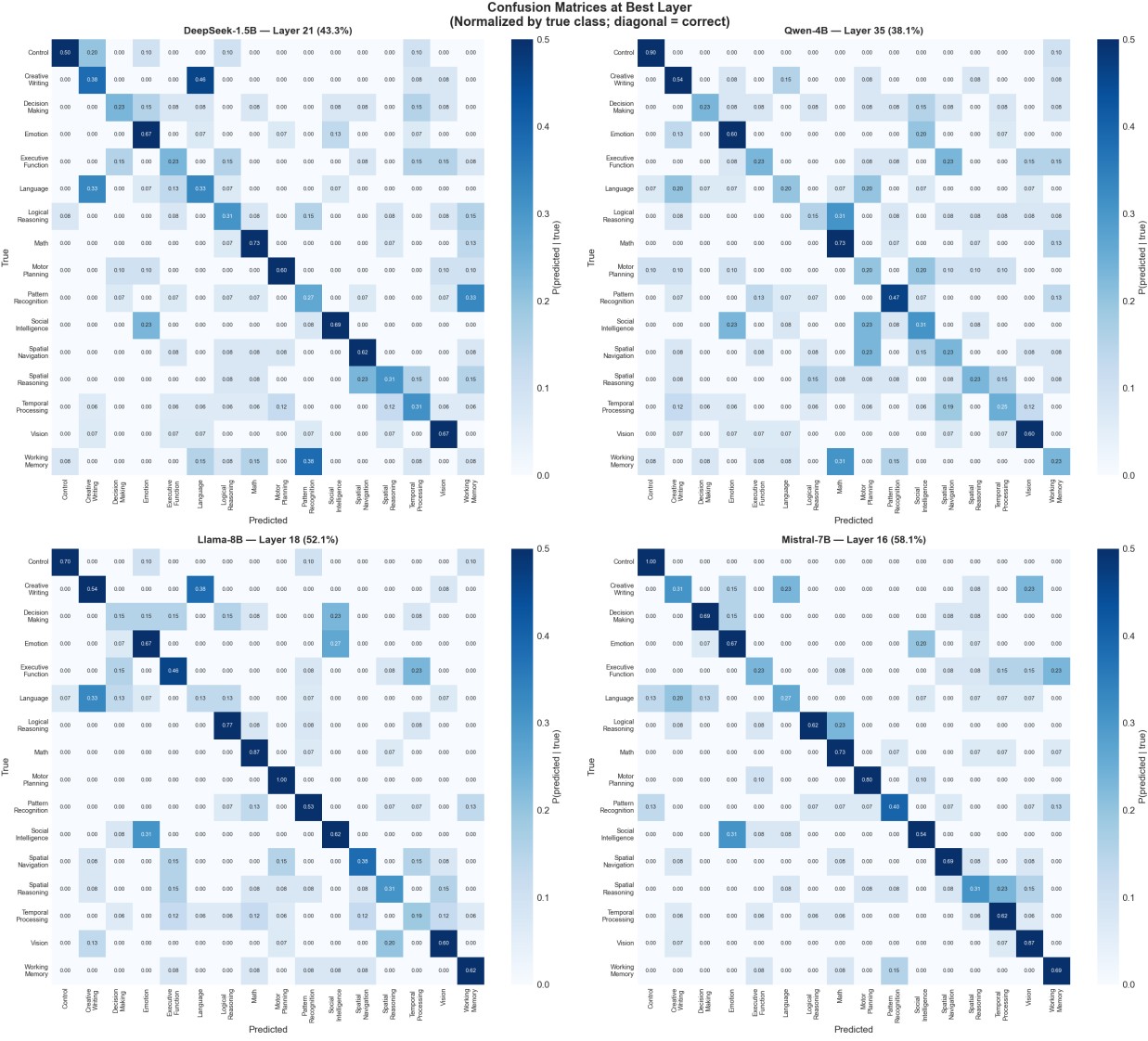

Figure 3: Confusion matrices for probe predictions (selected layers or models); rows true class, columns predicted.

## A.3 Information Dynamics

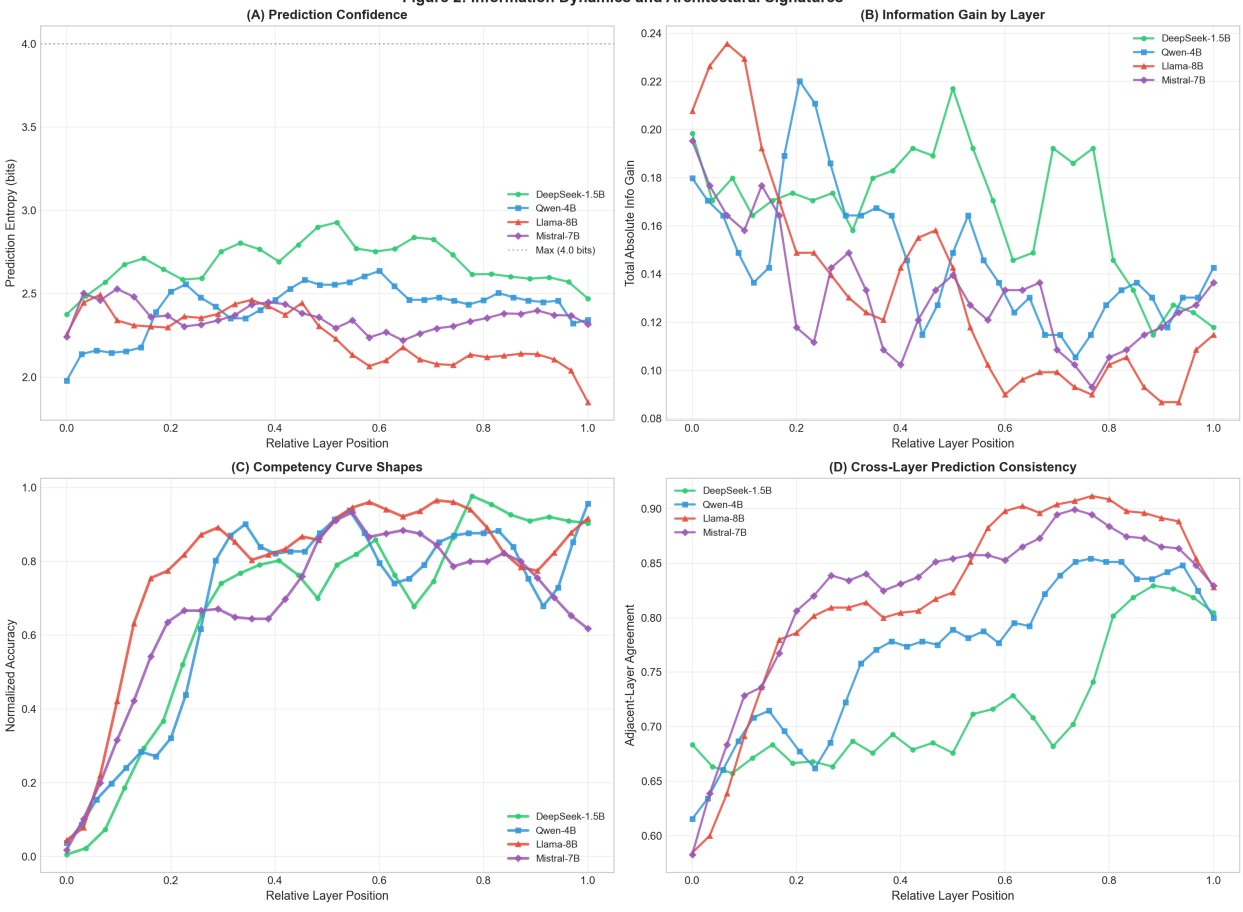

Figure 4: Information dynamics: (A) prediction entropy by layer, (B) information gain, (C) competency curve shapes, (D) cross-layer consistency.

## A.4 Cross-Architecture Efficiency

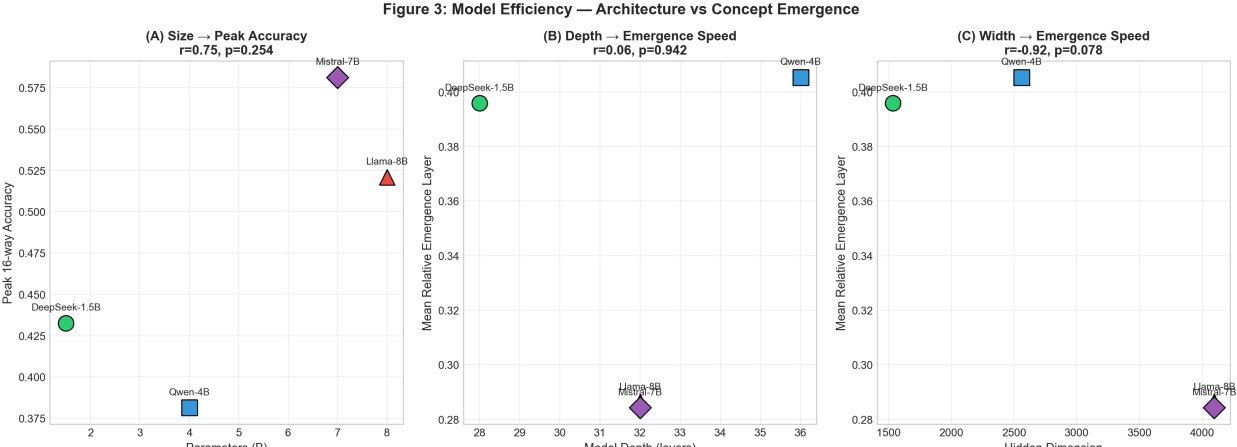

Figure 5: Efficiency vs. comprehension: emergence layer and accuracy by model; architectural differences in concept encoding speed.

## A.5 Representational Similarity

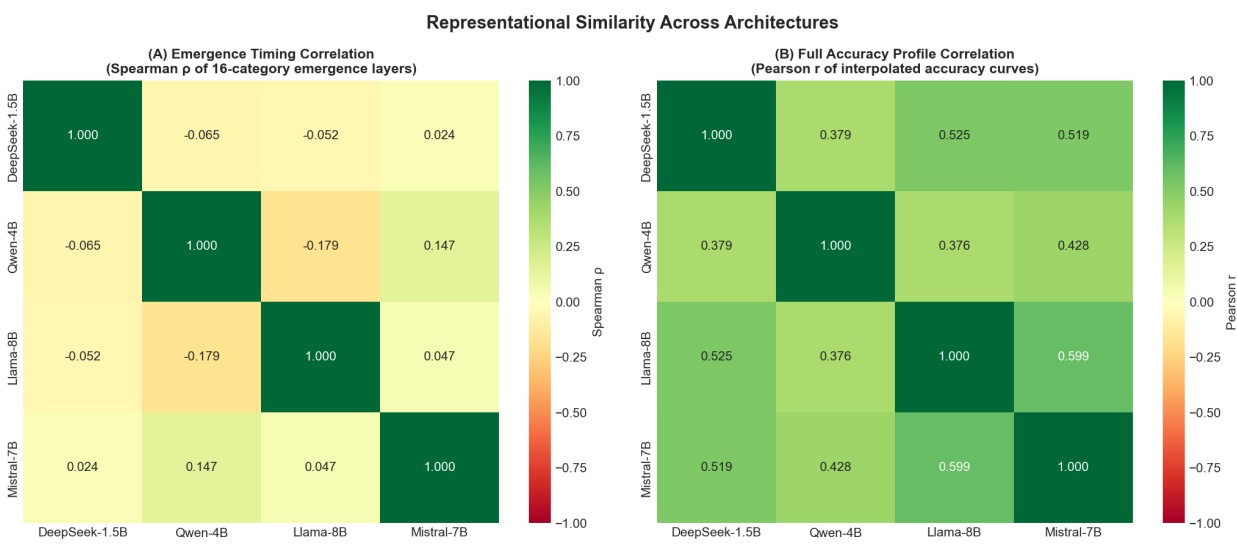

Figure 6: Cross-model representational similarity. **(A)** Spearman $\rho$ for pairwise emergence timing: all model pairs fall near zero ($-0.18$ to $0.15$), confirming that emergence *timing* is architecture-specific. **(B)** Pearson $r$ for full layer-accuracy profiles: moderate positive correlations ($r = 0.38$–$0.60$), strongest for the Llama–Mistral pair ($r = 0.60$), consistent with their shared hidden dimension and design.

## A.6   Category Clustering

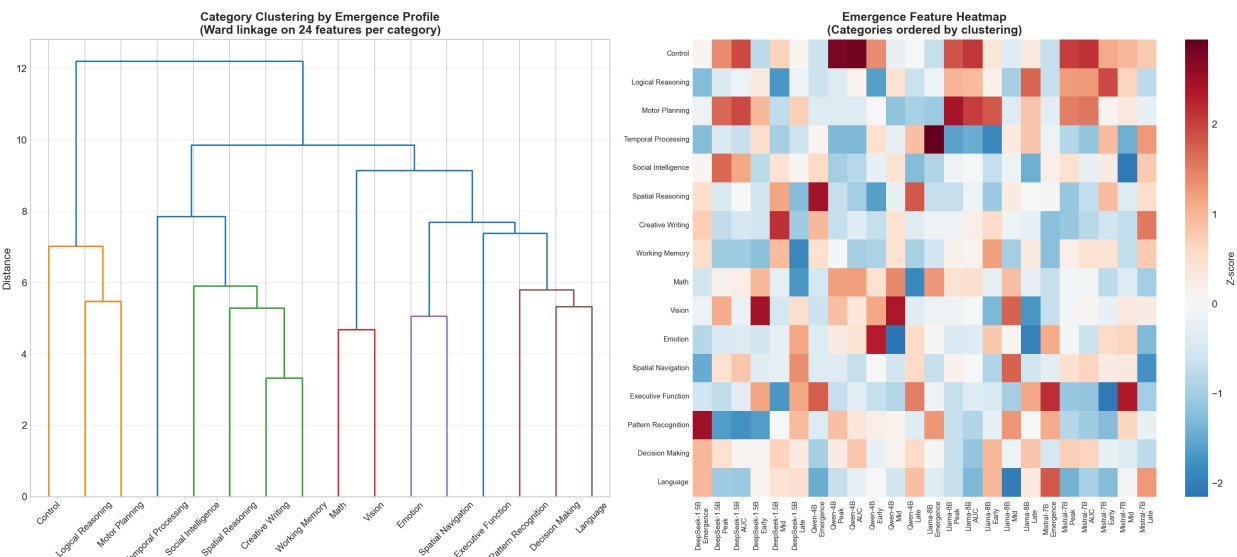

Figure 7: Hierarchical clustering (Ward linkage) of all 16 categories using 24-dimensional emergence feature vectors aggregated across models. Four natural groups emerge: early-structured (Control, Logical Reasoning, Motor Planning), late-variable (Creative Writing, Social Intelligence, Spatial Reasoning, Temporal Processing, Working Memory), unique-profile (Math, Vision), and mixed (remaining categories). Colors encode cluster assignment.

## A.7   SAE-Based Unsupervised Feature Discovery

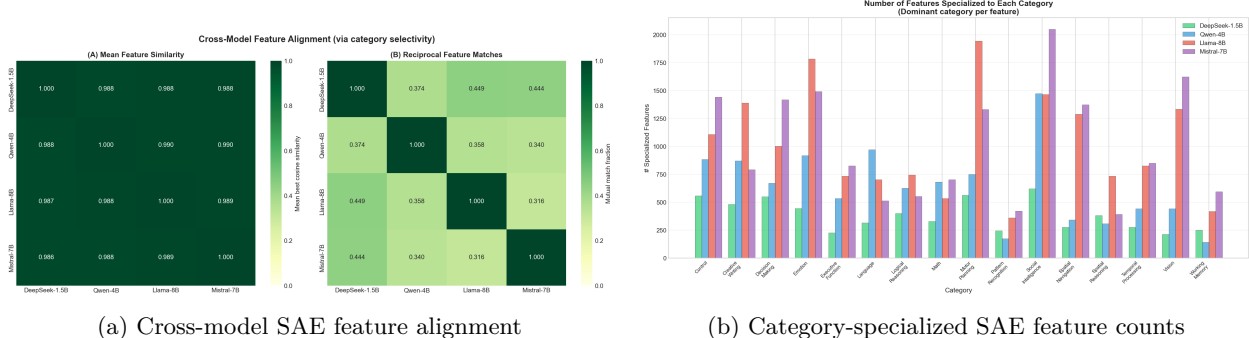

(a) Cross-model SAE feature alignment

(b) Category-specialized SAE feature counts

Figure 8: SAE-based unsupervised feature discovery. Panel (A) shows strong cross-model alignment in category-selectivity space. Panel (B) shows that all categories are represented by specialized features across architectures.

## B   Full Competency Heatmaps

Full competency heatmaps for all four models are provided in Figure 1 (main text). Each cell $(l, c)$ represents the probing accuracy for layer $l$ on category $c$, with green indicating high accuracy and red indicating low accuracy.

## C Per-Model Emergence Details

Table 8: Layer specialization summary per model.

| Metric | DeepSeek | Qwen | Llama | Mistral |
|---|---|---|---|---|
| Mean specialization score | 0.038 | 0.049 | 0.042 | 0.042 |
| Most specialized layer | 2 | 34 | 3 | 31 |
| Critical info-gain layer | 19 (70%) | 7 (20%) | 2 (6%) | 1 (3%) |
| Initial entropy (bits) | 2.376 | 1.979 | 2.257 | 2.241 |
| Final entropy (bits) | 2.472 | 2.343 | 1.850 | 2.317 |
| Entropy change | $+0.096$ | $+0.364$ | $-0.407$ | $+0.076$ |

## D Bootstrap Confidence Intervals

Full bootstrap results (1000 resamples) for emergence layers per category are shown in Figure 9. All key findings (universal emergence hierarchy, early-phase dominance, late-layer divergence) remain robust under resampling.

## E SAE Supplementary Diagnostics

This appendix extends Section 3.12 with additional diagnostics for feature inventory quality and training behavior. The same conclusions hold: feature dictionaries remain active, category-selective, and comparable across models.

The inventory view in Figure 10 is consistent with the main-text alignment analysis (Figure 8), providing an architectural comparison at the feature quality level in addition to category-selectivity structure.

## F Unsupervised Category Taxonomy

Figure 11 shows the unsupervised category map built from best-layer confusion profiles without using the supervised category labels. Despite being fully unsupervised, the map recovers meaningful pairings: Creative Writing clusters with Language, Logical Reasoning with Math, and Social Intelligence with Emotion. This agreement with the supervised probe confusions (Table 7) provides independent evidence that the confusion structure reflects genuine representational overlap rather than label-assignment artifacts.

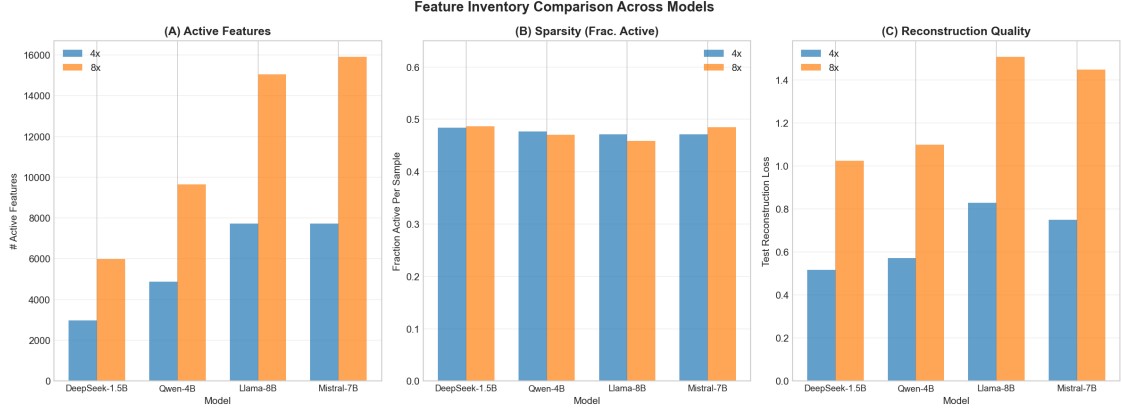

Figure 9: Bootstrap 95% confidence intervals for emergence layers; stability of emergence estimates.

Figure 10: SAE inventory diagnostics across models and expansion factors: mean active features per sample, sparsity (fraction active), and reconstruction loss.

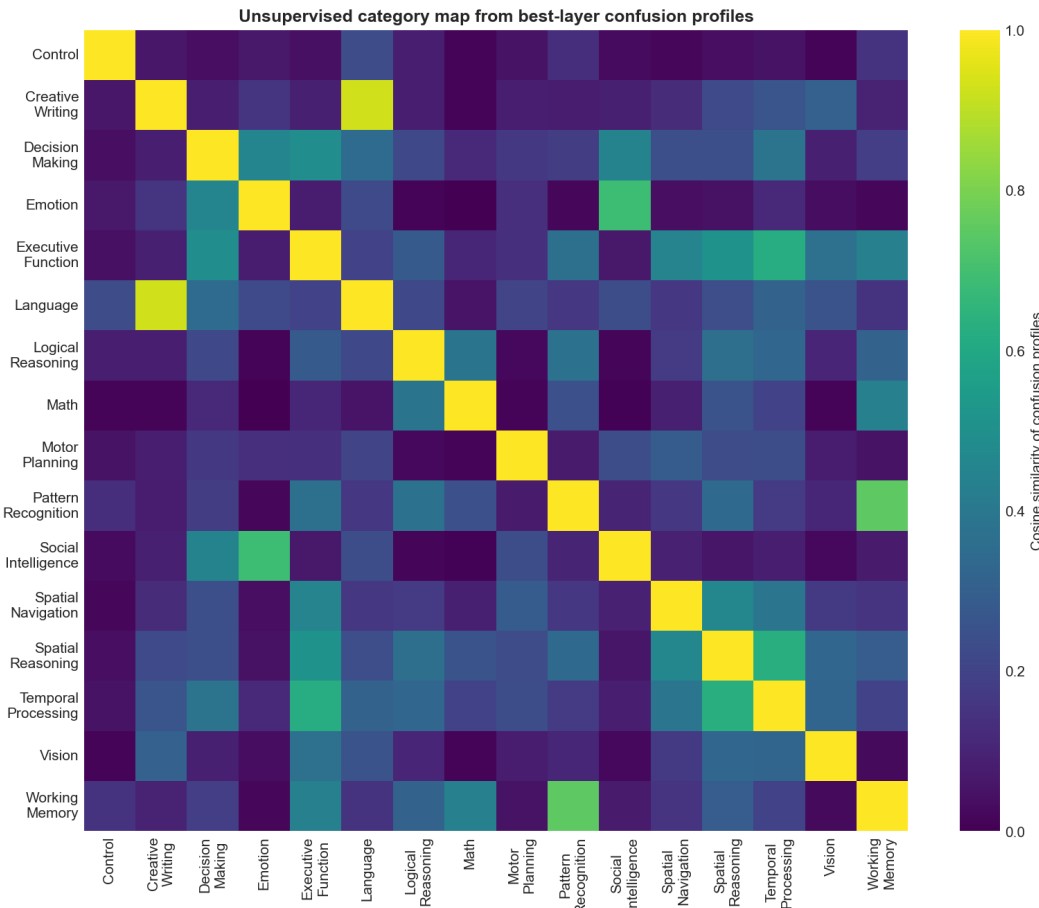

Figure 11: Unsupervised category map constructed from best-layer confusion profiles across all four models. Proximity indicates representational similarity. Recovered pairs (Creative Writing–Language, Logical Reasoning–Math, Social Intelligence–Emotion) match the supervised confusion structure from Table 7, providing cross-validation without supervision.

