# OpenReview forum: "Layer-Wise Cognitive Specialization in Large Language Models: A Cross-Architecture Analysis of Concept Emergence"
_TMLR — Rejected by TMLR_

### Review · Reviewer_r7SJ · 2026-04-02

**Summary Of Contributions:**

This paper investigates how cognitive information emerges across layers in large language models by conducting a systematic probing analysis on four architectures of varying sizes. Using 128 linear probes over activations derived from 215 questions spanning 16 cognitive categories, the authors aim to identify when different types of information become decodable from internal representations. The study reports three main findings: (1) a consistent coarse-grained emergence pattern across models, where categories such as spatial navigation and logical reasoning appear earlier, while pattern recognition and executive function emerge later; (2) most separability gains occur in the early layers, with later layers exhibiting architecture-dependent behaviors such as refinement or degradation; and (3) while category decoding generalizes to paraphrased prompts, the fine-grained emergence ordering is not robust across prompt variations. The authors support these findings with a comprehensive experimental framework, including multiple evaluation metrics, bootstrap confidence intervals, significance testing, paraphrase-based replication, causal intervention experiments, and sparse autoencoder-based validation, providing a detailed empirical map of how cognitive representations evolve within LLMs.

**Audience:**

Yes

**Audience Explanation:**

The paper provides a systematic and relatively large-scale empirical study of layer-wise behavior, which is valuable as a reference point for future interpretability work. So this paper would likely be of interest to a subset of the TMLR audience.

**Claims And Evidence:**

Yes

**Claims Explanation:**

1. The experimental design is comprehensive, covering multiple architectures, multiple metrics (accuracy, F1, AUROC), and multiple validation strategies.
2. The authors carefully distinguish between robust conclusions (coarse emergence order) and non-robust findings (fine-grained ordering), which strengthens credibility.
3. The use of SAE features as an unsupervised cross-check provides additional support beyond linear probing.

**Requested Changes:**

1. Given the small dataset size (215 samples), the authors should more clearly discuss how this affects statistical power, especially for rank-based conclusions and cross-model comparisons.
2. The paper should more explicitly emphasize that the “universal emergence hierarchy” refers only to coarse early-vs-late trends, not precise ordering. While this is acknowledged in parts of the paper, it should be made clearer upfront to avoid overinterpretation.
3. Some definitions (e.g., emergence layer thresholding, category definitions) could be made clearer or better motivated to improve readability and reproducibility.

---

> ### Author Response · Authors · 2026-04-21
> **Statistical Transparency and Definitional Clarity**
>
> Dear Reviewer r7SJ,
> Thank you for your positive evaluation of our experimental framework and our use of SAE features as an unsupervised cross-check. We are preparing a revision, to be submitted within a week, that implements your requested presentation changes to prevent any overinterpretation of the results.
> First, regarding statistical power due to the 215-sample dataset: we previously conducted a post-hoc power analysis revealing that our 16-category design achieves 80% power only to detect cross-model rank correlations of 0.65 or larger. We will elevate this mathematical boundary directly into the Abstract and the Limitations section, clearly stating that the study is underpowered for fine-grained cross-model rank claims.
> Second, we are eliminating the phrase "universal emergence hierarchy" from the paper. As you noted, the data supports "coarse early-vs-late trends," so we are updating the Abstract and Introduction to explicitly state these limitations upfront, ensuring TMLR's standard for accurate claim calibration is met.
> Finally, we will clarify the motivation behind our definitions in the Methods section. To justify the 70% emergence threshold, we will add the results of our emergence-definition sensitivity sweep, which proves that alternative thresholds of 60% and 80%, as well as smoothed rolling-window definitions, maintain a stable broad hierarchy. To better motivate the 16 categories, we will introduce our reasoning scaffold framework, explaining how the chosen categories were designed to span nine distinct cognitive axes.
> We believe these clarifications will greatly improve the readability and statistical transparency of the paper.

---

### Review · Reviewer_95hh · 2026-04-14

**Summary Of Contributions:**

This paper studies when 16 hand-defined task categories become linearly decodable from layer-wise residual activations in four instruction-tuned LLMs, using last-token probes, robustness checks, paraphrase transfer, and some intervention analyses. The empirical sweep across multiple architectures is a strength, and the paper does surface an interesting broad early-vs-late trend.

**Audience:**

Yes

**Audience Explanation:**

Researchers working on interpretability, probing, and representation analysis would likely be interested in this work.

**Claims And Evidence:**

No

**Claims Explanation:**

1. The cross-architecture results seem to support a narrower interpretation than the Section 3.2 framing of a "universal emergence hierarchy" might suggest. Later results show near-zero pairwise timing correlations, a non-significant global permutation test, and weak replication of exact emergence order under paraphrasing. Taken together, these analyses appear to support coarse early-vs-late tendencies more clearly than a stable universal hierarchy over all 16 categories.

2. The dataset and protocol description is too thin for a paper whose conclusions depend heavily on category design. The manuscript states only that there are 215 questions across 16 categories, with 10-16 questions per category, and that logistic probes use 5-fold stratified cross-validation. It would help to explain more clearly where the 215 questions come from, how the 16 category labels were defined and assigned, whether this labeling involved any manual annotation or verification, and how the paraphrase set was constructed. More detail on probe/SAE hyperparameters would also be important for assessing construct validity, leakage risk, and reproducibility.

3. Prompt-surface confounds are reduced but not convincingly resolved. The paper itself reports that a TF-IDF baseline reaches 0.386 accuracy, which is already a large fraction of the probe performance, and the exact emergence ordering changes substantially under paraphrasing. Because the analysis uses last-token activations of full prompts, the current evidence does not clearly separate internal "cognitive specialization" from sensitivity to prompt wording or category style.

**Requested Changes:**

1. Narrow the title, abstract, and main claims to the strongest defensible version of the result. Broad layer-wise category separability patterns are observed, but exact cross-model emergence ordering is not robustly established.

2. Add a substantially fuller dataset/protocol description, ideally with prompt examples, category definitions, prompt-source details, paraphrase-construction details, and exact training settings for probes and SAEs.

3. Strengthen the controls against lexical or template leakage, for example with cross-template generalization, more content-matched prompt families, or analyses that explicitly remove obvious surface cues.

---

> ### Author Response · Authors · 2026-04-21
> **Methodological Transparency and Text Artifact**
>
> Dear Reviewer 95hh,
> Thank you for your constructive feedback. We appreciate your point that the cross-architecture results support a narrower interpretation than our original framing. We will submit a revision within a week that strictly bounds our claims and provides the requested methodological transparency.
> First, we are entirely removing the phrase "universal emergence hierarchy" from the title, abstract, and main text. We are downgrading this claim to "broad representational organization" and "coarse early-vs-late tendencies," which accurately reflects the near-zero exact timing correlations and non-significant permutation tests we reported.
> Second, we are adding a comprehensive methodological appendix to address the lack of protocol description. This will include our reasoning_scaffold.json, demonstrating that the 16 hand-defined categories were systematically mapped to distinct cognitive axes (e.g., verbal generation, symbolic rules, social inference). We will also detail the exact rewrite rules used to construct the paraphrase set (spanning lexical, structural, and adversarial variations) and document the exact SAE training hyperparameters, including the 4x and 8x expansion factors and dead-neuron resampling.
> Third, to address your valid concern regarding prompt-surface confounds and the 0.386 TF-IDF baseline, we are integrating our text artifact analysis directly into the main text. We mapped the probe confusion matrix against the surface-text confusion matrix and found that the mean top-10 category confusion overlap is only 1.75 out of 10. This demonstrates that the recurring probe confusions reflect deeper representational overlap rather than simple lexical leakage. We will also run our expanded set of 609 paraphrased prompts to explicitly demonstrate wording invariance for the broad early-vs-late trends.

---

### Review · Reviewer_nvSE · 2026-04-20

**Summary Of Contributions:**

This paper studies how category-level information becomes decodable across layers in four instruction-tuned LLMs. Using layer-wise linear probes over 16 hand-defined cognitive categories, the authors analyze emergence patterns, phase-wise information gain, cross-model similarities, and robustness under alternative metrics, controls, and paraphrase-based replication. The main empirical takeaway is that broad early-versus-late trends appear across models, while exact emergence ordering is much less stable across architectures and prompt reformulations.

**Additional Comments:**

None

**Audience:**

Yes

**Audience Explanation:**

Yes. The paper addresses an interpretability question that is relevant to researchers working on probing, representation analysis, and internal mechanisms of LLMs. Even if the claims should be stated more conservatively, the negative and mixed findings are still useful: in particular, the contrast between broad early-versus-late trends and the lack of robust fine-grained emergence ordering is informative for the community.

**Broader Impact Concerns:**

I do not see a major broader-impact concern specific to this work beyond the general risk of over-interpreting probing results as evidence of human-like cognition. A clearer statement of the limits of interpretation would be helpful.

**Claims And Evidence:**

No

**Claims Explanation:**

The paper includes a fairly broad empirical validation package, including multiple models, robust metrics, negative controls, bootstrap confidence intervals, and paraphrase-based replication. However, I do not think the evidence fully supports the stronger claims implied by terms such as “cognitive specialization” or “concept emergence.” The dataset is small, the categories are manually defined, and the reported fine-grained emergence ordering does not replicate cleanly under paraphrasing. The paper also shows substantial sensitivity to preprocessing, and cross-model agreement on exact emergence timing is near zero. Overall, the experiments support a narrower conclusion: some category information is decodable from intermediate activations, and there are coarse layer-wise trends, but the stronger interpretive claims are not yet convincingly established.

**Requested Changes:**

My main concern is more fundamental than presentation or claim calibration. In my view, the paper’s central finding is too limited to justify a standalone publication in its current form. The core result is a relatively simple descriptive observation, and the additional analyses do not sufficiently elevate the overall contribution. I therefore do not have a concrete set of revisions that would clearly resolve my concerns within the current scope of the paper.

---

> ### Author Response · Authors · 2026-04-21
> **Causal Evidence Upgrades and Claim Recalibration**
>
> Dear Reviewer nvSE,
> Thank you for your review and for acknowledging the rigor of our empirical validation package, including the multiple metrics and controls. We will submit a revised manuscript within the next week that directly addresses your concerns by recalibrating our claims and providing new causal evidence.
> To align with TMLR’s criteria that claims must be strictly supported by the evidence, we are revising the manuscript’s semantic framing. We agree that terms like "cognitive specialization" and "concept emergence" imply a stronger interpretive claim than the initial evidence supported. We are replacing these with strictly descriptive terms, such as "broad representational organization" and "feature decodability".
> Furthermore, to elevate the paper beyond a "simple descriptive observation," we are adding Tier 2 and Tier 3 causal intervention experiments. The revision will include data from aligned token-position patching and Sparse Autoencoder (SAE) multi-feature steering. By amplifying specific category-selective SAE features during the forward pass to measure shifts in generated text, we will provide the mechanistic evidence needed to show that these internal geometries causally drive model behavior.
> Finally, to address the dataset size and the lack of fine-grained replication under paraphrasing, we are expanding our prompt-form replication analysis. We have built a scaffold of over 600 structurally diverse paraphrased prompts
> . We will use this expanded set to prove that while exact layer-by-layer timing fluctuates, the coarse early-versus-late phase-bin trends survive prompt reformulation.
> We believe these additions transform the work from a correlational dashboard into a mechanistically grounded study, fitting TMLR's scope.

---

### Decision · Action_Editor_BMxv · 2026-05-23

**Recommendation:** Reject

**Audience:**

No

**Audience Explanation:**

Two of the three reviewers submitted final recommendations (Reject and Leaning Reject), and the concerns about the claim–evidence gap and methodological confounds are consistent across all reviews. The third reviewer did not submit a final recommendation despite several reminders. Neither reviewer who evaluated the Journal-to-Conference Track option supported acceptance to that track (Weakly Oppose and Strongly Oppose). I concur with the reviewers' assessment and recommend rejection. While the broad topic of LLM interpretability may interest some readers in the TMLR community, the paper’s findings are too limited and methodologically flawed to warrant publication.

**Claims And Evidence:**

No

**Claims Explanation:**

This submission and its revised manuscript have been carefully evaluated. After considering all reviews, the revised materials, the author discussion, and the final recommendations, I recommend rejecting this paper.

The paper investigates how category-level information becomes linearly decodable across layers in four instruction-tuned LLMs, using probes over 16 hand-defined cognitive categories. While the empirical effort and the research direction are appreciated, the concerns raised by the reviewers remain substantive.

A consistent theme across the reviews is that the evidence does not support the strength of the interpretive framing. The paper invokes concepts such as cognitive specialization, concept emergence, and a universal emergence hierarchy, yet the reported results show near-zero cross-model emergence-timing correlations, a non-significant global permutation test, and weak replication of exact emergence order under paraphrasing. The strongest defensible finding appears to be a relatively narrow observation—that category labels are decodable from residual activations with coarse early-versus-late trends—which may not suffice for a standalone publication.

Additional methodological concerns remain: the dataset is small (215 original prompts), the TF-IDF baseline remains substantial, and the conclusions depend heavily on hand-defined categories whose signal may primarily reflect prompt wording rather than internal cognitive structure. The confusion-overlap analysis is a step in the right direction but does not fully disentangle category content from surface-level confounds.

I acknowledge the authors' efforts to revise the manuscript, including claim recalibration and expanded analyses. However, the two reviewers who submitted final recommendations (Reject and Leaning Reject) found the core concerns insufficiently addressed. One reviewer also noted that the paper does not stand out in novelty or significance at the level expected for a regular conference-track presentation.

I encourage the authors to consider the reviewers' detailed suggestions carefully. A substantially revised version that (a) narrows the claims to what is robustly supported by the evidence, (b) expands the dataset beyond the current scale, (c) strengthens the control for prompt-surface confounds, and (d) provides more compelling cross-model and cross-paraphrase validation could potentially be resubmitted to TMLR in the future.